# Eutopian and Dystopian Water Resource Systems Design and Operation—Three Irish Case Studies

## J. Philip O'Kane

School of Engineering, National University of Ireland, University College Cork and Ireland,
T12 K8AF Cork, Ireland; jpjokane@icloud.com

**Abstract:** The Harvard Water Program is more than sixty years old. It was directed by an academic Steering Committee consisting of the professors of Government and Political Science, Planning, Economics, and Water Engineering. In 2022 we would add to the notional Steering Committee the professors of Ecology, Sociology and Water Law, calling it the augmented Harvard eutopian approach to the design and operation of Water Resource Systems. We use the Greek word 'eu-topos' to mean 'a good place', figuratively speaking, and 'dys-topos' its antonym, 'not a good place'. By opposing eutopia and dystopia (latin forms) (Utopian literature begins with Thomas More's (1478–1535) fictional socio-political satire "Utopia", written in Latin and published in 1516: "*Libellus vere aureus, nec minus salutaris quam festivus, de optimo rei publicae statu deque nova insula Utopia*". "A little, true book, not less beneficial than enjoyable, about how things should be in a state and about the new island Utopia" [Wikipedia translation]. He coined the word 'utopia' from the Greek ou-topos meaning 'no place' or 'nowhere'. It was a pun-the almost identical Greek word eu-topos means 'a good place'), we pass judgement on three Irish case studies, in whole and in part. The first case study deals with the dystopian measurement of the land phase of the hydrological cycle. The system components are distributed among many government departments that see little need to cooperate, leading to proposition 1: A call for a new Water Law. The second case study deals with a project to restore a 200 km$^2$ polder landscape to its condition in 1957. The project came to the University with an hypothetical cause of the increased flooding and a tentative solution: dredge the Cashen estuary of its sand, speeding the flow of sluiced water to the sea, and the *status quo ante* would be restored. The first scientific innovation was the proof that restoration by dredging is impossible. Pumping is the only solution, but it raises disruptive questions that are not covered by Statute. The second important innovation was the discovery in the dynamic water balance, of large leakage into the polders, either around or between sluiced culverts, when the flap valves are nominally closed, impacting both their maintenance and minimization of pumping. Discussions on our findings ended in dystopian silence. Hence proposition 2: Moving towards eutopia may only be possible with a change in the Law. The third case study concerns the protection of Cork City from flooding: riverine, tidal and groundwater. The government's "emerging solution" consists of major physical intervention in the city centre, driven hard against local opposition, as the only possible solution. Two hydro-electric reservoirs upstream were largely ignored as part of a solution because the relevant Statute did not mandate their use for flood control. The Supreme Court has recently overturned this interpretation of the governing Statute. A new theory of flood control with a cascade of reservoirs, dams and weirs is the scientific innovation here. Once more these findings have been greeted by government with dystopian silence. Hence proposition 3: Re-open the design process to find several much better solutions, approximating a eutopian water world.

**Keywords:** eutopia; public administration; public policy making; water law; water resources; value for money; national water agencies; communication and silence; innovation; drainage; flooding; hydropower; propositions





## 1. Introduction

In 1966 the ground-breaking Harvard Water Program of the Graduate School of Public Administration (1955–1960) published the first in a series of books: "Design of water-resource systems: new techniques for relating economic objectives, engineering analysis, and governmental planning" reporting the results of a training and research programme directed by the Harvard academics: Arthur Maass (Government and Political Science), Maynard Hufschmidt (Planning), Robert Dorfman and Stephen Marglin (Economics), Harald A. Thomas, Jr., and Gordon Maskew Fair (Water Engineering) with senior employees of federal and state Water-Resource Agencies seconded to the project "in preparation for positions of greater responsibility in the public service" [1].

The Harvard research findings were organized in accordance with "the four steps of design". They are 1. identifying the objectives of design, 2. translating these objectives into design criteria, 3. using the criteria to devise plans for the development of specific water resources systems that fulfill the criteria in the highest degree, and 4. evaluating the consequences of the plans that have been developed, while making use of the digital computer whenever possible. This methodology is congruent with classical 'engineering-economic rationality': decisions are made between alternatives that address the fundamental questions: why do it at all, why do it this way, why do it now?

From the perspective of 2020, we would now insist on the participation of the professors of 1. Ecology, 2. Sociology, and 3. Water Law, in the research and training team. We call this the augmented Harvard design eutopia. Their presence would also challenge the importance given to the still-fashionable concept of System [1].

A Water Resource System is a water-object notionally divided into a set of constituent parts that interact with each other and with their environment, like a machine: consistently, purposefully, without contradiction, or illogicality, . . . . Furthermore, each alternative division of the water-object privileges the point of view of one stakeholder over others. As soon as we attach people to the water-object, it ceases to be a logically coherent Water Resource System becoming instead a people-messy Water Resource Complex. The former we may call a eutopian *terminus ad quem*, the latter a dystopian *terminus a quo*, termini on a spectrum of engineering rationality. The following case-studies illustrate this distinction and give rise to propositions that may ove the man-water dystopian Complex towards a eutopian water-world.

The place of systems analysis in managing water and the environment is essentially political and is the subject of an enquiry in a companion paper [2].

Since the Republic of Ireland is a member state of the European Union, national water policy is evolving to comply with the latest EU Water-Framework Directive https://ec.europa.eu/environment/water/water-framework/info/intro_en.htm, accessed on 1 June 2022, which in turn is a part of the environment framework https://environment.ec.europa.eu/index_en, accessed on 1 June 2022. Ireland's response to this challenge has been politically contested, and often delivered one government department or agency at a time, without an overarching Water Law.

Recently established water agencies with statutory authority are: the national water utility, Uisce Eireann—Irish Water [2], incorporated in July 2013 as a company under the Water Services Act 2013, An Coimisiún um Rialáil Fóntais—The Commission for Regulation of Utilities (CRU) 1999 and 2017 [3], and the advisory body, An Fóram Uisce—the National Water Forum [4], established in June 2018, under the Water Services Act 2017. This paper is not concerned with their individual difficulties and challenges, for example, the failure to implement the EU User Pays Principle (UPP) in domestic water supply, or whether UPP applies to flood protection. Rather, the case studies deliver a small number of propositions (theses or *Stellingen* NL) for better Public Policy Formation and Administration of a water world considered as a single integrated water resource complex. The propositions expressed here are concrete, arising from the case studies. Our notional augmented Harvard design eutopia is composed of relevant University Professors, and consequently are not representative of the people or of government.

The paper also presents a small number of local scientific innovations in case studies 2 and 3. They are not described in scientific or engineering detail.

## 2. Case Study 1

Fifty-eight years ago, the Republic of Ireland established "A National Institute for Physical Planning and Construction Research, 1964–1993 (http://isad.ie/units/view/id/437 (accessed on 1 June 2022); Environmental Research Unit 1988–1993, https://www.irishstatutebook.ie/eli/1993/si/409/made/en/print (accessed on 1 June 2022), https://www.irishstatutebook.ie/eli/1988/si/20/made/en/print (accessed on 1 June 2022)) (An Foras Forbartha—AFF)" with assistance from the United Nations Development Program. It reported to its parent Department (Ministry) of Local Government (and Environment) (DLGE). Each of the Institute's divisions had an advisory committee drawn from relevant central government departments, local authorities, business and commercial interests, and academia. The advisory committee for the Water Resources Division produced an inaugural report that was sent 'upstairs' to DLGE. The report was received with silence. Years later it was discovered that the cover had been stamped "ignore" before it had left the Institute (Minister James Dooge, personal communication, c. 1983). We may speculate, why?

There were two academic Systems Hydrologists (NUI Professors James Dooge and Eamonn Nash) on the advisory committee. Since measurement is the beginning of understanding, we may imagine great attention being paid to all the components of the land phase of the hydrological cycle: an atmosphere/vegetation/soil hydrological system of fields, tributaries, rivers, lakes, and estuaries, fed from surface, saturated and unsaturated subsurface runoff.

Unfortunately, the measurement of the hydrological components in this Systems View is the statutory responsibility of different government agencies and departments. For example, the Meteorological Office, Met Eireann, strives to meet the needs of transport (ships and aircraft), agriculture, fishing, and recreation, but not at sufficient spatio-temporal resolution for the identification of floods. A mere twenty-five rain gauges measure hourly rainfall at present (https://www.met.ie/climate/available-data/historical-data (accessed on 1 June 2022). The 9,500,000 individual hourly rainfall measurements in the National Climate Database for the period 1939 to 2017 imply fourteen gauges on average, compared to the current twenty-five of one per county, or one per $84,421/25 = 3376$ km$^2$ (a square of side 58 km). The "delivery of a high quality national flood forecasting service" foreseen in Met Eireann's Strategic Plan 2017–2027 will require a very big increase in the density of hourly rainfall gauges at key locations. This is not explicit in the plan. The Geological Survey attends to groundwater for water supply, water quality, and Karst flooding (https://www.gsi.ie/en-ie/programmes-and-projects/groundwater/activities/groundwater-flooding/Pages/What-is-groundwater-flooding.aspx (accessed on 1 June 2022)). The Office of Public Works measures river flow and surface water level in its role as the lead agency for flood alleviation, ignoring groundwater (https://www.gov.ie/en/organisation/office-of-public-works/ (accessed on 1 June 2022)). The Department of Agriculture relies on Met Eireann for estimates of evapo-transpiration and soil moisture in the root zone of crops, for mechanized harvesting, spraying, irrigation, and drainage. The Electricity Supply Board (https://www.esb.ie/what-we-do/generation-and-trading/hydrometric-information (accessed on 1 June 2022)) measures flow and surface water level wherever it generates hydropower, but has said in the High Court it would like to offload this burden. (The High Court judgment (Judge Max Barrett); https://www.rte.ie/documents/news/ucc-v-the-electricty-supply-board.pdf (accessed on 1 June 2022). "An additional four tributary river gauging stations were installed during the 1980s. . . . Nowadays such measurements are not necessary and the gauges work as water level gauges only. Latterly, ESB has had contact with the OPW and local authorities with a view to transferring these stations across to them." Page 87.) The recently established national water utility, Irish Water, measures the abstraction and distribution of water to industry, business, and households.

No one measures routinely sediment concentration and flux, or adsorbed chemical species, at a high enough frequency to detect pulses of pollution. The Environmental Protection Agency, the inheritor in 1993 of part of AFF, tries to bring order to this dystopian world in its Hydrology Summary Bulletins.

Met Eireann has indicated that the number of WMO-standard rain-gauges (daily average rainfall) may have to be reduced due to cutbacks. Off-the-shelf battery-powered meteorological stations connected by Wi-Fi to the cloud every few minutes are now sufficiently cheap (https://www.netatmo.com/en-gb/weather (accessed on 1 June 2022)) that dozens of citizens have installed them across Ireland. Measurements are posted to the cloud but lack national Quality Assurance and dissemination. This is an opportunity for agencies of the state to cooperate through innovation in partnership with the citizenry, and with the science laboratories of primary, secondary, and tertiary schools.

The national UNESCO IHP-ICID Committee hosted by the Office of Public Works (OPW; https://hydrologyireland.ie/about-the-national-committees/ (accessed on 1 June 2022)) and related Inter-departmental Committees are not sufficiently powerful to drive a eutopian Systems View of the hydrological cycle.

*Proposition 1*

A new Water Law for integrated water resource management in Ireland could deliver a cost-effective eutopian solution to the measurement problem. There are model Water Laws; for example, the Portuguese Water Law (https://files.dre.pt/1s/2005/12/249a00/72807310.pdf (accessed on 1 June 2022)) of the 1980s.

## 3. Case Study 2

In 1997, the Chief Civil Engineer of the Office of Public Works (OPW) requested assistance from the Department of Civil and Environmental Engineering, (NUI, Cork; UCC), in solving a flooding problem in the rural parts of the Lower Feale River and Cashen Estuary, a tributary of the outer Shannon, the largest river in Ireland. The Arterial Drainage Division of OPW had avoided closure during the deep economic retrenchment of the late 1980s, because it was required by statute to maintain previous arterial drainage schemes, but it was shorn of staff and lacked modern expertise.

The River Feale and Cashen estuary scheme was constructed by the commissioners of public works between 1951 and 1957 under the 1945 arterial drainage act. This low-lying coastal area is approximately 200 square kilometers in extent. A pre-existing polder landscape was re-engineered with an extensive system of river embankments and back-drains dewatering the polders through sluiced culverts (pipes with flap-valves through the embankments) that open at low tide to allow the accumulated rain and upland runoff to escape to the estuary and the sea. The fields in each polder lie below sea level for a fraction of each tidal cycle. When there is a flood on the Feale river, the culverts may be unable to open, even at low tide; the back drains overflow, and the fields are inundated.

The number, elevation, and positioning of sluiced culverts in the embankments surrounding the fifteen polders were the crucial design decisions. The most important design tool was a laboratory hydraulic model of the embanked rivers and tributaries. The final design also specified a trapezoidal channel dredged down the Cashen estuary and lined with bitumen (?) to take the sluiced rainwater from the polders to the sea as fast as possible in accordance with the theory of arterial drainage.

The scheme was judged to have been a success. The farmers were able to grow cereals, high in commercial value, but with the passage of time, this became more difficult as the frequency of inundation increased. Eventually, marsh plants returned to the fields in the polders, and cattle grazing became the main farming activity. At the same time, storms eroded the lined channel in the estuary, and eventually, no trace of it could be seen. Consequently, the project came to the University in 1997 with a hypothetical cause of the increased flooding and a tentative solution: dredge the Cashen estuary of its sand, speeding the flow of sluiced water to the sea, and the *status quo ante* would be restored. But for how

long, before a repetition would be required? Sand and gravel merchants, suppliers to the construction industry, were already standing by to assist with future excavation.

The University (NUI, Cork; UCC) agreed to accept the challenge on eutopian terms use of:

1. The best instrumentation. e.g., the German Aerospace DLR-HRSC 'Mission to Mars' stereoscopic multi-spectral camera to survey the ground DEM on a 200 km$^2$ grid of <50 cm cells, to an accuracy of ~15 cm in all three directions, and georeferenced to the National grid (www.dlr.de (accessed on 1 June 2022)).
2. The best simulation software. e.g., the Danish Hydraulic Institute's suite of Mike modelling systems (www.dhigroup.com (accessed on 1 June 2022)).
3. A three/four-year stipend for a Ph.D. student, and
4. Reasonable expenses.

The goal was to make and calibrate a high-resolution computer model of the entire polder system of the lower Feale and Cashen estuary containing all its tributaries, bridges, embankments, tidal barrages, sluiced culverts, backdrains, and agricultural fields: a hydroinformatic world in which hundreds of alternative solutions could be tested using measurements made during historic flood events.

Six examples illustrate the local scientific innovation arising from this two-part systems study; the first two examples are simple, the third is eutopian, the fourth and fifth are partly eutopian, and the sixth is dystopian.

1. The difference between measured and modeled water level at several points in the Feale river every 20 min during one whole year was ~10 cm or less, except in March and September, when the difference increased to an unacceptable ~50 cm. The error was in the data, not in the model, and originated from the OPW procedure for reducing data from autographic recorders. The procedure introduces negligible errors when there are no tidal components in the signal; but the opposite is the case in a tidal river or estuary, at those times when the Winter–Summer hour-change occurs. We had identified a lacuna in OPW's scientific reduction of data.
2. Changing the number, location, and elevation of sluiced culverts in the model was found to yield at most a 10 to 15% improvement in the alleviation of flooding in the polders. Such a small improvement is not worthwhile, since the farmers would see expensive activity with no observable benefit. At a working meeting with OPW engineers, we pointed to one place, where the model said a second culvert would produce a relatively large improvement. Two days later we were informed that a second culvert was already present, but the engineer in charge of its installation had not updated the files from which we were working. The scientific credibility of the model soared. We had identified a lacuna in the files.
3. We call Social Calibration of models the presentation of dynamic flood maps (for particular flood events) to stakeholders with personal experience of these events. They are asked to identify their farm, fields, buildings, houses, etc., and to pronounce on the accuracy of the simulation. The test group was the OPW maintenance staff of the Feale drainage scheme. They were shown animations and asked to fill in a questionnaire (with maps) commenting on the spatial extent of the most recent large flood. This feedback led to improvements in the model and a further increase in its credibility.
4. The most important scientific innovation of the first part of the Feale systems study [3] was the contradiction of the hypothesis, which had arrived with the project, that unplugging the Cashen estuary would restore the fields in the North Kerry polders to their original condition in 1957.

A comparison of maps of the Lower Feale peatlands, from Alexander Nimmo's bog survey of c.1810 (http://sources.nli.ie/Record/MS_UR_050570 (accessed on 1 June 2022)), with subsequent surveys by the Ordnance Survey of Ireland in the nineteenth and twentieth centuries, and with the DLR-HRSC-A-derived DEM (digital elevation model), shows the

long-term contraction and decline of the peat landscape due to drainage for agriculture and the harvesting of peat for fuel.

The computer model was used to examine a very large number of arterial dredging schemes from the mouth to beyond the confluence with its two most important lowland tributaries. Dredging is OPW's métier for getting the flood to the sea as fast as possible, by increasing the hydraulic 'conveyance' of the channel. So why does dredging not work in the Lower Feale?

The scientific study of drained peat and fen landscapes in England, the Netherlands, and North Germany, has shown that drainage for increased agricultural output causes the peat land to sink, due to the loss of buoyant uplift in the now-aerated root zone of crops, and the bio-oxidation of the peat component of the aerated soil. As the land sinks, the sluiced culverts gradually cease to work, not because there may be more sand on the seaside of the embankments (although that has an influence), but because of the drop in the surface level of the fields. In these circumstances, the only solution is to pump from the back-drains.

An empirical formula from continental European studies on the rate of subsidence of drained peat soils allowed us to estimate the 1957 surface level of the fields in the polders. The model (without dredging) showed almost no flooding, in agreement with the original scheme. At present, the Feale polders correspond roughly to the year 1500 on the timeline of the Netherlands when wind-mill-driven pumps were first introduced to dewater back-drains that could no longer be emptied with sluiced culverts.

But this story creates problems for OPW. The 1945 Arterial Drainage Act (http://www. acts.ie/en.act.1995.0014.1.html (accessed on 1 June 2022); https://www.irishstatutebook.ie/ eli/1995/act/14/enacted/en/html (accessed on 1 June 2022)) requires the OPW to "return the scheme to the required standard in compliance with OPW's statutory requirements". (Quotation from an OPW call for bids (May 2022) entitled "Cashen Estuary Pumping Infrastructure—cover note". The note fails to mention that drainage has already caused the peat landscape to sink.) It is physically impossible to return the scheme to its original state, a situation not anticipated by the 1945 Act. What to do?

If the pumps maintain the water level in the polder fields as close as possible to the lip of the back and field drains, there will be no flooding and subsidence will be minimized, but the water table will continue to lie close to the surface of the fields preventing increased agricultural output, the ultimate goal of arterial drainage. If farmers demand an aerated root zone for the crops they choose to grow, the subsidence will continue, the sluiced culverts will gradually cease to function, and the frequency of pumping will increase from year to year.

In the Netherlands, the set point for on/off pumping is determined by a political process at the local level. The pumping of the 10 km$^2$ polder at the North Slob (https: //en.wikipedia.org/wiki/North_Slob (accessed on 1 June 2022)) in Wexford, may be the sole Irish example. The farmers in that polder own and operate the pumping station and choose their own set-points.

Furthermore, particular crops may be agreed each year with the National Parks and Wild Life Service (https://www.wexfordwildfowlreserve.ie (accessed on 1 June 2022)) to foster a wintering ground of international importance for a number of migratory waterfowl, including in particular Greenland White-fronted Geese and Brent Geese, as well as Bewick Swans and Wigeon. OPW might study this case to help answer the disruptive questions:

➤ What does Sustainable Development of the peaty Feale polders mean? (Look to Wexford and the Netherlands!)
➤ Since restoration of the 1957 scheme is impossible, how should the on/off set point for pumping be lowered from year to year? (Look to Wexford and the Netherlands!)
➤ Increasing the storage in the back drains by widening them, thereby losing agricultural land, reduces pumping. What's the trade-off?
➤ When will maintenance of the peaty Feale resource become uneconomic? Legal statute requires 'Value for Money'.

These questions can only be answered if they are posed in a larger framework, preferably a eutopian framework with inputs from agronomists, economists, ecologists, and the Kerry Cooperative Society, which holds shares in its multi-national food giant, Kerry Foods, one-time owner of the North Slob and its EU-CAP milk quota (https://kerryco-op.com (accessed on 1 June 2022); https://www.kerry.com (accessed on 1 June 2022); https://leestrand.ie/the-company/ (accessed on 1 June 2022); https://www.independent.ie/regionals/newrossstandard/news/extensive-farm-on-north-slob-sells-for-57-million-27494503.html (accessed on 1 June 2022)).

OPW agreed to join with the University in a follow-on project (part two of the Feate system study), modelling and measuring two adjacent polders, one with a new OPW pumping station and the other acting as a control. A set of automatic rain gauges, piezometers and water level recorders, and an eddy-correlation meteorological station, measured the response of the water table in both polders to (1) the ambient hydro-meteorology, Feale river flow and tide, and (2) the simultaneous operation of the sluiced culverts and pumping station. We registered the site as a CEOP Hydrology Reference Site: Sleeven Polder, lower Feale River basin, County Kerry, Ireland (drainage area ~7.5 km$^2$).

The HRSC-A camera made a second overflight. Ground-penetrating radar and electrical resistivity surveys provided estimates of soil properties in the root zone to calibrate a Mike-SHE model of the surface and subsurface hydrology, which could be used to drive a growth model for agricultural crops. A vertical stainless-steel post in a plastic tube was driven into the ground at the eddy-correlation station to measure long-term subsidence, copying the Holme Fen Posts in the drained Whittlesey Meer near Peterborough in England (https://www.greatfen.org.uk/about-great-fen/heritage/holme-fen-posts (accessed on 1 June 2022)).

The most important scientific innovation of the follow-on study [4,5] was the discovery, in the dynamic water balance, of large leakage into the polders, either around or between sluiced culverts, when the flap valves were nominally closed. This has important implications for both the maintenance and design of the culverts and embankments, and consequently for minimizing pumping. Unfortunately, we did not include electrical conductivity probes in the suite of instruments. Measuring the changing salinity on both sides of the embankments might identify the location of the leakage at sluiced culverts and/or along back drains. Confirmation with a Rhodamine B dye-test might also be tried with a controlled field experiment on a culvert.

Minister Deenihan, TD for North Kerry, requested a public lecture in his constituency on the project findings and on alternative courses of action. Attendance was large, the questions penetrating and the audience most attentive. The session ended with a question from us to the farmers and the wider community: What is the most appropriate use of the polders? flax, hemp, cereals, silage, grazing, forestry, . . . or hydraulic cultivation of cranberries, and perhaps Dutch-style tourism of an historic polder landscape with the ruined Old Court of Lixnaw at its heart? Meneer Van de Leur, who immigrated to nearby Kilrush in Co. Clare in the early seventeenth century may have been the first to advise the Fitzmaurice, Barons Lixnaw, and Earls of Kerry, on polder technology.

Repeated requests to OPW for a meeting with agronomists from the Department of Agriculture and for information on the set-point of the pumping controller were met with silence. We had outstayed our welcome in OPW, a public institution with a long-standing internal separation of power between legal and executive officers and engineers, a recipe for dystopia.

*Proposition 2*

Internal, and therefore invisible, divisions in government agencies are correlated with statutory authority and administrative policy. Moving towards eutopia may only be possible with a change in the Law. The third case study provides a final illustration of this.

**4. Case Study 3**

*4.1. Preamble*

The OPW is an old and venerable part of our Public Administration. At present, it reports to the Department of Public Expenditure and Reform (DPER; https://www.gov.ie/en/organisation/department-of-public-expenditure-and-reform/# (accessed on 1 June 2022)).

Following the financial crash of 2008, the government retrenched to restore the financial sovereignty of the state. The posts of City Engineer, County Engineer, and Chief Engineer (*inter alia*) were stood down. Private sector consulting firms were presumed to be capable of filling the gaps.

I imagine the situation: Managers of engineering services in the Public Service would write the specifications for the design and operation of engineering projects, call for bids, and ensure blind competition in their evaluation, in some cases assisted by other consultants also selected by competition. These managers would require a minimum level of expertise to understand the problems, pose the questions, and engage in a techno-economic conversation with private sector consultants and the political process.

In keeping with the 'new managerialism', OPW now has a Customer Charter, not a Citizen's Charter. It begins with a mission statement: "the mission of the Office of Public Works is to use our experience and expertise to fulfill our role and responsibilities with effective, sustainable and innovative services to the public and to our clients with competence, dedication, professionalism and integrity. . . . Our ethos is client focus, timely delivery and value for money." The Customer Action Plan 2017–2019 says "Our customers are Government, other Departments, Offices and Agencies, and the public (no capital letter). As a member of the public, as a Citizen of the Republic, we may comment "in a fair . . . open . . . and respectful manner".

The following case study shows how the 'new managerialism' prevents cost-effective innovation, moving government away from eutopia towards dystopia.

*4.2. The Great Flooding of Cork City*

In November 2009, almost the entire center of Cork City was flooded with water released from two hydro-electric dams upstream on the river Lee, the largest release since the construction of the two reservoirs in the 1950s. The semi-state Electricity Supply Board (ESB) is the dam operator. Several hundred domestic and commercial buildings, including the most recent flag-ship buildings of the University (UCC), were flooded. There was no loss of life.

The center of Cork is also flooded from the sea during high-spring tides and storm surges that overtop the open quays of this ancient maritime city. Parts of the low-lying central island are also subject to groundwater flooding.

City Hall (https://www.corkcity.ie/en/ (accessed on 1 June 2022)) asked the Office of Public Works (OPW), as the 'lead' government agency for dealing with floods, to find an engineering solution that would prevent a repetition of the flooding at an acceptably low level of risk.

*4.3. OPW's "Emerging Solution" Methodology*

OPW established a steering committee, appointed consultants, and applied its putative design methodology:

1   Define the problem in engineering hydrologic and hydraulic terms.
2   Search for constraints that narrow the set of alternatives to a single solution, and call it the "emerging solution".
3   Carry out a multi-criteria impact assessment of the "emerging solution" to demonstrate that all constraints have been satisfied.
4   Carry out a cost–benefit analysis to show that it delivers 'value for money'.
5   Publicly defend the "emerging solution" as the only possible solution.

There are no publicly available minutes of meetings of the Steering Committee to confirm (1) the absence from the Committee of OPW and City Hall Heritage Officers for the Built Environment, and (2) the following speculative ex-post rationalization.

"The Irish Government Economic and Evaluation Service (https://igees.gov.ie (accessed on 1 June 2022)) (IGEES, established 2012) is an integrated cross-Government service to enhance the role of economics and value for money analysis in public policy making." If only one "emerging solution" is considered, it is impossible to demonstrate "value for money".

The "emerging solution" methodology is dystopian because it treats constraints as infinitely rigid and consequently each has an infinite shadow price. Comparing finite shadow prices shows where a trial design can be improved. Economists define "value for money" with the three-Es (economy, efficiency, effectiveness) micro-economic theory of Resource Allocation, taught to every student of Business and Economics. It is the eutopian Harvard merger of engineering and economics.

### 4.4. OPW's "Emerging Solution" for Protecting Cork

The "emerging solution" for protecting Cork is a plan to wall in the Central Island and canalize the North and South Channels with 14 km of 1.4 m high embankments, walls, walls with gaps, and 'demountables', together with many groundwater pumps, at a cost of ~ EUR 140 million.

The pumping system may be designed as if the Central Island were a single large building on soft ground, but there are hundreds of old and new buildings threatened with differential settlement and cracking. As of 2017, not a single measurement of the groundwater response to river flood and tide had been made between the dams and the city and across the Central Island.

The public was invited to City Hall to admire the photomontages of the increasingly detailed "emerging solution" in three "have-your-say-days", exercises in faux participative democracy.

All attempts to broaden the discussion of alternatives to include interventions away from the city, such as changes to the dams upstream and a tidal barrier downstream, failed. The case for such engineering alternatives rests on my discovery by computer simulation that an alternative operating procedure for the dams would have eliminated the flood of November 2009 entirely. Consequently, 1.4 m high walls and embankments are not required to protect the city from the river.

### 4.5. Flooding from the Sea and Groundwater

The problem of flooding from the sea and from groundwater remains to be addressed. Each of the three cities, Belfast, Dublin, and Cork, on the east coast of Ireland, has considered a tidal barrier to protect against tidal flooding. The Lagan Weir and Tidal Barrier in Belfast were built in the early 1990s at a fifth of the cost (adjusted for inflation) of the "emerging solution" for Cork. It also has the very great advantage of controlling both water level and water quality in a polluted section of the Lagan as part of a new city development.

The key question is where to locate the barrier. If it is too close to the city, the river may flood the city during the period when the barrier is closed. The OPW consultants produced a report dismissing the bigger tidal barrier promoted by the citizens of "Save Cork City" as far too expensive, technically impossible, and environmentally intrusive. The width of the tidal Lee at Tivoli is the same as the width of the Lagan Weir. A smaller barrier is well worth examining. It is not a piece of exotic engineering, but it would require OPW and its consultants to innovate.

Theory shows that two hydroelectric reservoirs (with new fusegates on one of them), a new reservoir at Drumcara, the Lee Fields washlands between the dams and the city, and a tidal barrier with a large pump, located between Tivoli and Blackrock, when conjunctively operated with my new control procedure, can protect the city with minimal interference with hydro-electricity generation. The total cost could be half the cost of the "emerging solu-

tion". https://hydrologyireland.ie/wp-content/uploads/2021/12/01-OKane-P-Poster-01-OOK-211104.pdf (accessed on 1 June 2022)

The "emerging solution" therefore represents, in my view, a substantial waste of public funds, the conversion of the city center of Cork to a building site for the second time in a generation, the destruction of the character of an eight-hundred-year-old maritime port with open quays, and lost opportunities to actively control water level around the Central Island and further downstream. https://hydrologyireland.ie/wp-content/uploads/2021/12/02-OKane-P-Poster-01-OOK-211104.pdf (accessed on 1 June 2022)

This critique has been greeted with silence by the ESB, OPW, and City Hall. We may speculate why: the infinitely rigid legal constraint, recently found by the Supreme Court to be elastic.

*4.6. The Legal Constraint*

In response to the 2009 flood, the University instituted legal proceedings in the Irish courts seeking to recover flood damages from the ESB (https://www.irishstatutebook.ie/eli/1945/act/12/enacted/en/print.html (accessed on 1 June 2022)). The six-hundred-page judgment of the High Court found in favor of the University assessing contributory negligence at 40% (UCC): 60% (ESB; The High Court Judge Max Barrett; https://www.rte.ie/documents/news/ucc-v-the-electricty-supply-board.pdf (accessed on 1 June 2022). "A steering-group was established in March 2010 to drive forward the process for implementation of flood risk management measures for the lower Lee as envisioned in the draft plan. This steering group comprises the OPW, Cork City/County Council and ESB. It was decided to commission consultants to undertake the design, planning and construction supervision of proposed measures and to look at environmental issues." Pages 152–153).

The ESB appealed the judgment to the Court of Appeal. It overturned the judgment of the lower court declaring the University to be entirely liable for the damages (The Court of Appeal, The President, Irvine J., Whelan J. between [2016 No. 92] University College Cork—National University of Ireland and Electricity Supply Board Judgment of the Court delivered by the President on 20 March 2018. https://www.courts.ie/acc/alfresco/21708bf7-34cd-4dcb-8ba2-b3f4a83ce552/2018_IECA_82_1.pdf/pdf#view=fitH (accessed on 1 June 2022)), since the Act of Parliament under which the ESB operates the dams does not mandate flood-control. The University appealed the judgment to the Supreme Court, which found against the ESB (The Supreme Court University College Cork—National University of Ireland Plaintiff/Appellant and The Electricity Supply Board Defendant/Respondent Joint judgment of Mr. Justice Clarke, Chief Justice, and Mr. Justice MacMenamin delivered the 1 July 2020. https://www.courts.ie/acc/alfresco/3dddd234-f5ef-4977-8acf-8def4b794afc/2020_IESC_38%20Clarke%20CJ,%20MacMenamin%20J%20(Unapproved).pdf/pdf#view=fitH (accessed on 1 June 2022); "16.2 For the reasons set out earlier in this judgment, we have concluded that the approach identified in more recent United Kingdom case law, which analyzes liability on the basis of a "do no harm" approach, is to be preferred to the more traditional consideration which differentiated between acts of commission or acts of omission. However, we have also identified that there can be exceptions to the "do no harm" rule such that a duty of care may arise, in certain limited circumstances, to confer a benefit. 16.3 We consider that one such exception arises where a party is in a special position of control enabling them to prevent harm being caused by a danger independently arising. While we consider that the special level of control in question does not necessarily have to arise from the existence of a legal power, we are satisfied that it must be substantial and not tangential. We also suggest that it must be shown that there is a reasonable relationship between any burden which would arise from imposing such a duty of care and the potential benefits to those who may be saved from the danger in question. In addition, we are satisfied that it is necessary to ensure that it is possible to define the duty of care in question with a sufficient, but not absolute, level of precision so as to avoid imposing a burden which is impermissibly vague and imprecise." Page 88.)

The Court declared that the ESB had a duty to confer a flood-control benefit on the City "in certain [unspecified] circumstances", damages and costs to be decided by the lower courts.

After many years of litigation (as of Spring 2022) the legal process has not yet concluded. The decision of the Irish Supreme Court may be of interest to countries that follow the Common Law tradition as courts grapple with the question of whether it is lawful to separate flood control from the generation of hydropower.

But the Dáil (Parliament) could simply change the Law making explicit the multiple uses of the reservoirs that have arisen since their construction.

The United Nations Development Program and the UN Executive Agencies make a distinction between problem-orientated science and the application of appropriate technology: first the science, then the technical solution. The Office of Public Works does not make this distinction, outsourcing much design work to the private sector, especially engineering consultants who may innovate abroad but not at home. Innovation is mandated by the OPW Customer Charter.

**5. Proposition 3—Eutopian**

The project to protect Cork from pluvial, riverine, tidal, and groundwater flooding should be reopened with a multi-disciplinary steering committee (the augmented Harvard eutopian design approach), separating science and engineering, politics and Law, public participation and political representation, in the engineering, economic, ecological, and socio-cultural evaluation of innovative alternatives.

**6. Conclusions**

*6.1. Proposition 1*

A new Water Law for integrated water resource management in Ireland could deliver a cost-effective eutopian solution to the measurement problem.

*6.2. Proposition 2*

Internal, and therefore invisible, divisions in government agencies are correlated with statutory authority and administrative policy. Moving towards eutopia may only be possible with a change in the Law.

*6.3. Proposition 3*

Re-open the design process to find several much better solutions to flooding in Cork, approximating a eutopian water world.

A thread of bureaucratic silence runs through all three case studies. We await the day when Customer Charters will refer to Jonathan Lynn and Anthony Jays' [6]:

Three Varieties of Civil Service Silence, Discreet, Stubborn, and Courageous:

○ "Discreet Silence is the silence when they do not want to tell you the facts.
○ Stubborn Silence is the silence when they do not intend to take any action.
○ Courageous Silence is the silence when you catch them out and they haven't a leg to stand on. They imply that they could vindicate themselves completely if only they were free to tell all, but they are too honorable to do so.

How difficult it is to forthrightly tell the truth, the whole truth—the devil is in the detail.

**Funding:** This research received no external funding.

**Informed Consent Statement:** Not applicable.

**Data Availability Statement:** Not applicable.

**Acknowledgments:** I thank all my colleagues, graduate students, and public servants who contributed to these stories over many years. Errors of commission and omission remain with me.

**Conflicts of Interest:** The authors declare no conflict of interest.

## Notes

1    https://en.wikipedia.org/wiki/Shared_vision_planning, accessed on 1 June 2022.
2    https://www.water.ie, accessed on 1 June 2022.
3    https://www.cru.ie/home/about-cru/, accessed on 1 June 2022, "The Commission for Regulation of Utilities (CRU) is Ireland's independent energy and water regulator. The CRU was originally established as the Commission for Energy Regulation (CER) in 1999. The CER changed its name to the CRU in 2017 to better reflect the expanded powers and functions of the organisation. The CRU has a wide range of economic, customer protection and safety responsibilities in energy and water. The CRU's mission is to protect the public interest in Water, Energy and Energy Safety" but apparently not Water Safety (flooding, water quality, recreation).
4    https://thewaterforum.ie/what-we-do/#what-we-do, accessed on 1 June 2022.

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
