# Peer review of "Eutopian and Dystopian Water Resource Systems Design and Operation—Three Irish Case Studies"

_hydrology, doi:10.3390/hydrology9090159_

Round 1

Author Response

Thank you for your comments. Please see the reply.

Reviewer 2 Report

The current version of your manuscript is not a scientific communication. The target should be the international reader and at least the method should be presented to be applicable to other situation. Therefore my overall recommendation is to reject manuscript from publication.

Author Response

Thank you for your comments . Please see the reply.

Reviewer 3 Report

The paper is entitled 'Eutopian and Dystopian Water Resource Systems Design & Operation - three Irish case studies'. It represents the author's opinion on the design and exploitation of water resources based on 3 different cases in Ireland. In all three discussed cases, there is a problem of proper management of water resources, appropriate legal regulations that will correspond to the current hydrological conditions and problems, and financing of investments related to water resources management. The problem of outsourcing studies to external companies is quite common. This is due to the lack of proper personnel - good engineering consultants have their own design companies and the lack of cooperation between scientists from the university and the water resources administration. In my opinion, the topic addressed in the opinion would be a good example of a broader scientific article that shows the gaps in the existing water resource management system. This problem affects not only Ireland but many other European countries. To make the submitted opinion clearer, I propose to insert a short paragraph describing the current state of water resources management in Ireland.

Author Response

Dear Reviewer #2, I thank you for your comments. I have inserted a short paragraph describing the current state of water resources management in Ireland. It’s highlighted in red in the uploaded latest version of the paper. The endnotes to the paper contain the links supporting the statements in the red paragraph. Other small changes/additions to the original paper are also in red. Here is the additional red paragraph. 

Since the Republic of Ireland is a member state of the European Union, national water policy is evolving to comply with the latest EU Water-Framework Directive https://ec.europa.eu/environment/water/water-framework/info/intro_en.htm, which in turn is a part of the environment framework https://environment.ec.europa.eu/index_en. Ireland’s response to this challenge has been politically contested, and often delivered one government department or agency at a time, without an overarching Water Law.

Recently established water agencies with statutory authority are: the national water utility, Uisce Eireann – Irish Water[i], incorporated in July 2013 as a company under the Water Services Act 2013, An Coimisiún um Rialáil Fóntais – The Commission for Regulation of Utilities (CRU) 1999 and 2017[ii], and the advisory body, An Fóram Uisce – the National Water Forum[iii],  established in June 2018, under the Water Services Act 2017. This paper is not concerned with their individual difficulties and challenges, for example, the failure to implement the EU User Pays Principle in domestic water supply, or in flood protection. Rather the case studies deliver a small number of propositions (theses or Stellingen NL) for better Public Administration of a water world considered as a single integrated water resource complex. The propositions expressed here are concrete, arising from the case studies. Our notional augmented Harvard design eutopia is composed of relevant University Professors, and consequently are not representative of the people or government.

The paper also presents a small number of local scientific innovations in case studies 2 and 3. They are not described in scientific or engineering detail.

[i] https://www.water.ie

[ii] https://www.cru.ie/home/about-cru/ “The Commission for Regulation of Utilities (CRU) is Ireland’s independent energy and water regulator. The CRU was originally established as the Commission for Energy Regulation (CER) in 1999. The CER changed its name to the CRU in 2017 to better reflect the expanded powers and functions of the organisation. The CRU has a wide range of economic, customer protection and safety responsibilities in energy and water. The CRU’s mission is to protect the public interest in Water, Energy and Energy Safety” but not Water Safety (flooding).

[iii] https://thewaterforum.ie/what-we-do/#what-we-do

Yours Sincerely,

Philip O'Kane

Round 2

Reviewer 2 Report

I maintain my previous opinion.